# Validation of IOTA-ADNEX Model in Discriminating Characteristics of Adnexal Masses: A Comparison with Subjective Assessment

**DOI:** 10.3390/jcm9062010

**Published:** 2020-06-26

**Authors:** Soo Young Jeong, Byung Kwan Park, Yoo Young Lee, Tae-Joong Kim

**Affiliations:** 1Department of Obstetrics and Gynecology, Samsung Medical Center, Sungkyunkwan University School of Medicine, Seoul 06351, Korea; ohora_87@naver.com (S.Y.J.); yooyoung.lee@samsung.com (Y.Y.L.); 2Department of Radiology, Samsung Medical Center, Sungkyunkwan University School of Medicine, Seoul 06351, Korea

**Keywords:** ultrasonography, ovarian tumor, IOTA-ADNEX model, subjective assessment

## Abstract

(1) Background: The aim of this study is to compare the IOTA-ADNEX (international ovarian tumor analysis–assessment of different neoplasias in the adnexa) model with gynecologic experts in differentiating ovarian diseases. (2) Methods: All participants in this prospective study underwent ultrasonography (US) equipped with the IOTA-ADNEX^TM^ model and subjective assessment by a sonographic expert. Receiver operating characteristic (ROC) curves were also generated to compare overall accuracies. The optimal cut-off value of the ADNEX model for excluding benign diseases was calculated. (3) Results: Fifty-nine participants were eligible: 54 and 5 underwent surgery and follow-up computed tomography (CT), respectively. Benign and malignant diseases were confirmed in 49 (83.1%) and 10 (16.9%) participants, respectively. The specificity of the ADNEX model was 0.816 (95% confidence interval (CI): 0.680–0.912) in all participants and 0.795 (95% CI, 0.647–0.902) in the surgical group. The area under the ROC curve of the ADNEX model (0.924) was not significantly different from that of subjective assessment (0.953 in all participants, 0.951 in the surgical group; *p* = 0.391 in all participants, *p* = 0.407 in the surgical group). The optimal cut-off point using the ADNEX model was 47.3%, with a specificity of 0.977 (95% CI: 0.880–0.999). (4) Conclusions: The IOTA-ADNEX model is equal to gynecologic US experts in excluding benign ovarian tumors. Subsequently, being familiar with this US software may help gynecologic beginners to reduce unnecessary surgery.

## 1. Introduction

Gynecologic ultrasonography (US) is a useful tool for identifying the presence of ovarian mass, differentiating between benign and malignant tumors, and determining the treatment plan of ovarian tumors [1]. There are two important reasons to use US examination in differentiating ovarian tumors from benign to malignancy. First, precise differentiation of benign and malignant tumors helps clinicians to know what to do: observation or surgery, a surgery type, laparotomy or laparoscopy or other additional studies, tumor marker, computed tomography (CT), or magnetic resonance imaging (MRI) [2]. Second, ovarian malignancy is not so common among gynecologic tumors, but it is a fatal disease with high recurrence rates [3]. Most patients with ovarian cancer are diagnosed at an advanced stage, with a below-30% 5-year survival rate. However, the 5-year overall survival in early-stage cancer is as high as 92.4% [4], and thus early detection using US improves a patient’s survival rate. This may provide the patient with the most optimal follow-up or treatment with the most benefit at the least cost.

Several scoring models, such as the risk of malignancy index (RMI) and the risk of ovarian malignancy algorithm (ROMA), have been developed for the differentiation of adnexal masses [5,6,7]. Since 2005, the international ovarian tumor analysis (IOTA) group has presented other risk predictive models with logistic regression (LR1, LR2) [8,9] and sonographic characteristics (simple rules) [10,11]. The IOTA group demonstrated that these predictive models have better diagnostic performance than preexisting systems [12]. In 2014, a new model with better performance, the assessment of different neoplasias in the adnexa (ADNEX) model, was developed in this group [13]. This model uses three clinical features and six US features to predict the malignancy risk of adnexal masses. Through various external validation studies, the discriminating performance of ovarian tumors with the IOTA-ADNEX model has been better than other existing models [14,15,16,17,18,19]. However, there were rare prospective studies comparing this model and gynecologic experts in terms of characterizing ovary tumors. Our hypothesis is that the IOTA-ADNEX model is not different from the subjective analysis of experienced experts in differentiating benign and malignant ovary disease.

The aim of this study is to compare the IOTA-ADNEX model with gynecologic experts in differentiating benign and malignant ovary diseases.

## 2. Materials and Methods

### 2.1. Patient Selection and Study Design

This prospective pilot study was performed at a single institute (Samsung Medical Center, Seoul, Korea) between 28 March and 19 July 2019. This research was approved by the institutional review board (IRB No. 2018-12-034) of Samsung Medical Center. Written informed consent was obtained from each participant. The calculation of the sample size was determined by our statistical team. Under the hypothesis that the difference between the AUC of IOTA-ADNEX and subjective assessment is greater than the non-inferiority margin, the difference was calculated as 0.03 on average, and the non-inferiority limit was assumed to be −0.01. The minimal number of subjects satisfying the statistical power of 80% and a significance level of 5% was 56, and at least 62 patients were required when considering the drop-out rate of 10%.

All participants underwent transvaginal grayscale and color Doppler US examination to identify adnexal masses. The inclusion and exclusion criteria are listed in Figure 1. Peripheral venous blood was sampled to measure CA-125 as a tumor marker for ovarian tumors. If both the IOTA-ADNEX model and subjective assessment suggested a benign disease, abdomen and pelvic CT was scanned 3–4 months later. If either suggested ovarian tumors requiring surgery, surgery was performed within 6 months to confirm histologic examination. For histologic diagnosis of ovary diseases, a pathologist followed the World Health Organization International Classification of Ovarian Tumors and the International Federation of Gynecologists and Obstetricians (FIGO) 2012 classification in case of malignancy.

### 2.2. Ultrasound Examination

The equipment was a HERA W10 ultrasound system (SAMSUNG MEDISON Co., Ltd., Gyeonggi-do, Korea) with an endovaginal EA2-11B probe and IOTA-ADNEX^TM^. One gynecologic radiologist (B.K.P.), who had more than 20 years’ experience of gynecologic US, evaluated the ovary tumors. When ovary masses were identified on both sides, the more complex or larger mass was chosen for analysis.

Ovary masses were described according to terms and definitions of IOTA. Color Doppler US was applied and the scores were determined according to the blood flow: 1 = no blood flow, 2 = minimal blood flow, 3 = moderate blood flow, and 4 = marked blood flow. Subjective assessment classified ovarian masses into five scales: benign, probably benign, intermediate, probably malignant, and malignant [20].

The IOTA-ADNEX model needs the input of three clinical variables, including age (years), serum CA-125 level (U/mL), and type of center (oncology center/other hospital), and the measurement of six sonographic variables, including the maximal diameter of the lesion (mm), proportion of solid tissue (%), number of papillary projections (0/1/2/3/>3), more than 10 cyst locules, acoustic shadow, and ascites. The proportion of solid tissue is defined as the ratio of the maximal diameter of the largest solid portion to the maximal diameter of the lesion. In the type of center, an “oncology center” is defined as a tertiary clinical center with specific gynecologic oncology, and the rate of ovarian malignancy in oncology centers and other hospitals are usually “22~66%” and “below 30%”. By inserting these values in ultrasound equipment, the probability of benign and malignant risk is calculated, and the patient risk is expressed as a relative risk (RR) in comparison with the baseline risk with the prescribed formula. It represented five categories—benign, borderline tumors, stage I invasive, stage II-IV invasive ovarian cancer, and secondary metastatic cancer—and highlighted markers for large values of RR. The cut-off point of malignancy risk was 10%, and borderline tumors were included in the malignancy group. All principles related to the ADNEX model were followed by the practical guidelines [13,21].

### 2.3. Statistical Analysis

The descriptive statistics include means (range) for continuous variables and numbers (percentage) for categorical variables (ultrasound predictors and IOTA-ADNEX results). Clinical data were compared between benign and malignancy by χ^2^ or Fisher’s exact tests for categorical variables, and Student’s t- or the Wilcoxon rank-sum tests for continuous variables.

The diagnostic performance of IOTA-ADNEX models was obtained to compare with that of subjective assessments in differentiating benign ovarian tumors from malignancies by calculating specificity.

Receiver-operating characteristic (ROC) curves were analyzed for the IOTA-ADNEX model and subjective assessment and the areas under the curve (AUCs) were calculated. In order to compare the AUCs between two methods, a nonparametric approach for two correlated AUCs, which was proposed by Delong, was applied [22].

Optimal cut-off values were obtained to discriminate ovarian malignancy using the ADNEX model with the Youden index method. The Youden index is a measure for summarizing the performance of a diagnostic test. This measure was first introduced to medical literature by Youden [23].

Statistical analyses were performed with a commercially available software (SAS Institute, Cary, NC, USA). A *p*-value of less than 0.05 was considered as statistical significance.

## 3. Results

Consecutive 62 participants were screened, but three were excluded because of withdrawn consent (*n* = 3). The final cohort was 59 participants: 54 (surgical group) and 5 (nonsurgical group) underwent surgical intervention and follow-up with CT images, respectively. Participants’ demographics and US features are shown in Table 1. The mean age was 45 (range, 20–71) years and the mean CA-125 level was 43.4 U/mL (range, 2–672 U/mL). In terms of age and CA-125, there were significant differences between benign and malignant ovary tumors (*p*-values = 0.001 and 0.005).

Of 59 participants, 14 (23.7%) had bilateral tumors. The mean mass diameter was 65.5 mm (17–200 mm) and that of the solid component was 16.2 mm (0–86 mm). Ten participants (16.9%) had more than three papillary projections, and six (10.2%) had more than 10 cyst locules. Acoustic shadow and ascites were shown in 10 (16.9%) and 5 (8.5%) participants, respectively. Nineteen (32.2%) and 26 (46.43%) participants had solid and multilocular tumors, respectively. Sixteen patients (27.1%) had blood flow in Color Doppler. US features of benign tumors included a unilateral mass without papillary projection or blood flow. In contrast, US features of malignancies included ascites, solid portions, papillary projection, or blood flow.

Of 59 participants, 49 (83.1%) had benign ovarian masses and 10 (16.9%) had borderline or malignant tumors. Of the surgical group (*n* = 54), endometrioma (33.3%), mature cystic teratoma (14.8%), and serous cystadenoma (12.8%) were common in benign ovarian tumors (Table 2). Two borderline tumors of mucinous type (3.7%) and eight malignant tumors (14.9%) were identified in histologic exams (Table 2).

### 3.1. Diagnostic Performance of IOTA-ADNEX Models

For validation of the IOTA model, the specificity was calculated with 10% of cut-off value for differentiating ovarian tumors. This model had a specificity of 0.816 (95% CI: 0.680–0.912) in all participants and 0.795 (95% CI: 0.647–0.902) in the surgical group. In order to show that the specificity of the ADNEX model using the HERA W10 ultrasonography was not the same as that that was measured by another US device, z-test was applied, and there was a significant difference (*p* = 0.005 in all participants and *p* = 0.017 in the surgical group) [16]. The sensitivity of the IOTA-ADNEX model was 0.9 (95% CI: 0.555–0.998) in all participants and the surgical group.

### 3.2. ADNEX Model vs. Subjective Assessment

The AUC of the IOTA-ADNEX model was 0.924 (95% CI: 0.786–1.0) in all participants or the surgical group. The AUC of the expert’s subjective analysis was 0.953 (95% CI: 0.878–1.0) in all participants and 0.951 (95% CI: 0.874–1.0) in the surgical group. When comparing two AUCs with a nonparametric approach, there were no significant differences between the IOTA-ADNEX model and the expert’s subjective assessment (Figure 2; *p* = 0.391 in all participants and *p* = 0.407 in the surgical group).

### 3.3. Optimal Cut-Off

This study identified the optimal cut-off point of discriminating ovarian malignancy using the ADNEX model with HERA W10 at 90% sensitivity. The optimal cut-off point determined by the Youden index method in all participants was 47.3%, with a specificity of 0.980 (95% CI: 0.892–0.999). A similar result was shown in the surgical group (optimal cut-off value 47.3% with a specificity of 0.977 (95% CI: 0.880–0.999)). These values were higher than the original value of 10%.

We calculated the diagnostic performance (sensitivity, specificity, PPV, NPV, LR+, LR-, accuracy) at 5%, 10%, 15%, and the optimal cut-off point (Table 3). The specificity of the cut-off points of 5%, 10%, and 15% in both groups was not different, but there was a significant difference between the specificity of original (10%) and optimal (47.3%) cut-off points (*p* = 0.005).

## 4. Discussion

Our study shows that the IOTA-ADNEX model is relatively good at characterizing ovary masses, but also provides high specificity in estimating the risk of malignant ovary tumors. The diagnostic accuracy of the IOTA-ADNEX model was not different from that of subjective assessment by an expert (B.K.P.) who had performed gynecologic USs for more than 20 years at the gynecologic cancer center. As a result, our gynecologists (T.J.K. and Y.Y.L.) had strong confidence in determining how to make a surgical plan because so many ovary masses were preoperatively considered as benign tumors using this model.

Validation studies using the IOTA-ADNEX model have tried to find out the optimal cut-off value of malignancy risk [16]. The Youden index method suggested that the optimal cut-off point in the surgical group was 47.3%, with a specificity of 0.980 (95% CI: 0.892–0.999). Their results are very similar to those in our surgical group, showing that the optimal cut-off value was 47.3% with a specificity of 0.977 (95% CI: 0.880–0.999). These values were higher than the original values of 10%. When using this optimal cut-off value in the IOTA-ADNEX model, gynecologists can show better performance in identifying benign ovarian tumors with higher specificity and avoid unnecessary surgery in benign or probably benign cases, which are preoperatively diagnosed with this model.

RMI is frequently used in distinguishing ovarian tumors and was introduced in national guidelines such as the Royal College of Obstetricians and Gynaecologist (RCOG) guidelines. However, in a recent review article, RMI had poor diagnostic performance when compared to other mathematical models or sonographic-guided predictive models, such as SR or LR2, especially with low sensitivity, [16,24,25]. In addition, several studies have demonstrated that the RMI or ROMA is inferior to subjective assessment in discriminating benign from malignant adnexal masses [26,27].

Several studies have shown the advantage of the IOTA-ADNEX model in discriminating ovarian tumors in comparison with other methods, including subjective assessment. The common result was that the ADNEX model has high accuracy in diagnosing characteristics of ovarian tumors with high negative predictive value and exclusion of malignancy. Additionally, the ADNEX model has performed well in discriminating stage II-IV ovarian malignancies from other tumors [14,16,18].

Serum CA-125 has been most frequently used as a screening test of epithelial ovary tumors since it was developed in 1981 [28]. However, this tumor marker provides relatively poor specificity because it also increases in many benign conditions [29]. To overcome this limitation, many gynecologists utilize imaging studies, including US, CT, or MRI [30,31]. Most of all, US is a primary imaging modality, and CT or MRI is a problem-solving one for those who have inconclusive US results. A systematic review has shown that sonography alone gives great results in determining the characteristics of ovarian tumors and benign or borderline/malignancies preoperatively compared with using MRI or CT/PET-CT [32].

Our study had some limitations. First, the strength of this study was prospective, but the size of population was relatively small. We need further investigations to determine whether there is any difference regarding the different expertise and differentiation between benign tumors and different categories of malignancy. Second, the follow-up period was relatively short. The nonsurgical group had only three months for follow-up using CT. Longer follow-up is necessary to confirm benign ovary tumors. Third, one participant was false-negative in both the IOTA-ADNEX model and subjective assessment. The sonographic results were “probably benign” as a subjective assessment of an expert and “1%” as an overall malignancy risk in the IOTA-ADNEX model. However, the histologic diagnosis was a mucinous borderline tumor, in which a very small cancer focus was detected in the cystic mass. Fifth, the IOTA-ADNEX model is not available in transabdominal US. Therefore, women without sexual contact do not undergo this model. Finally, this model requires the input of sonographic parameters, which will presumably be dependent on the competence/experience of the sonographer. We definitely need to investigate if it is easy for beginners to use. However, in this study, the involved radiologist took a very short time to become familiar with the input, even though he had never used it before.

## 5. Conclusions

The IOTA-ADNEX model is a promising US software in precisely differentiating benign and malignant ovary tumors. It is equal to gynecologic US experts in terms of preoperatively characterizing ovary tumors. Particularly, the high specificity of this model may contribute to detecting benign tumors. Therefore, being familiar with this US software may help gynecologic beginners to reduce unnecessary surgery.

## Figures and Tables

**Figure 1 jcm-09-02010-f001:**
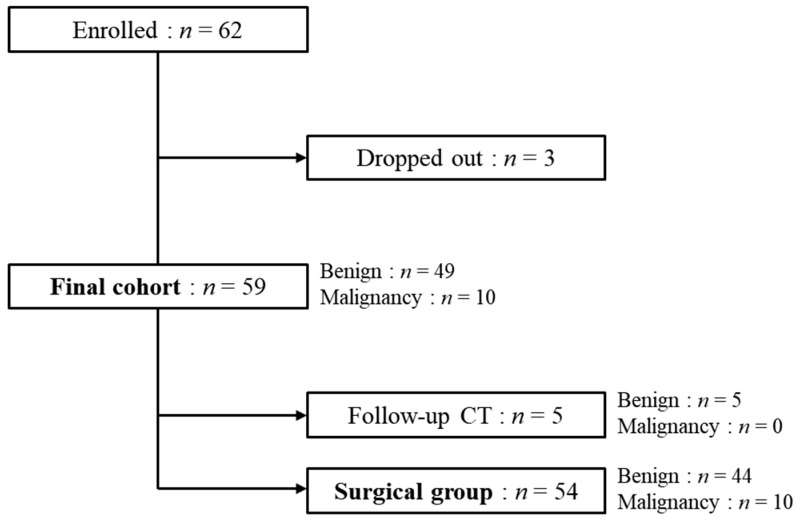
Flow chart of the study population.

**Figure 2 jcm-09-02010-f002:**
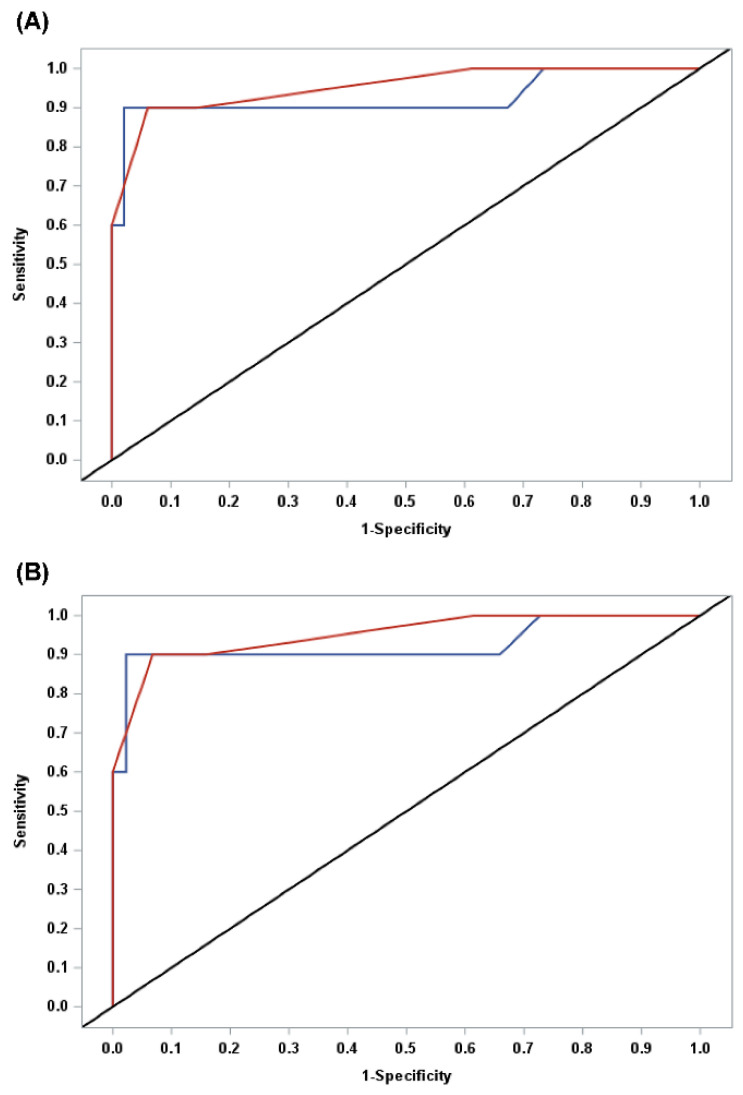
Receiver operating characteristic (ROC) curves of overall malignancy risk in the ADNEX (assessment of different neoplasias in the adnexa) model (blue) and subjective assessment (red) in (**A**) all participants and (**B**) surgical group.

**Table 1 jcm-09-02010-t001:** Participants’ demographics and ultrasonography (US) findings.

	Benign (*n* = 49)	Malignant (*n* = 10)	Total (*n* = 59)	*p*-value
Clinical characteristics				
age, yr (range)	42 (20~68)	59 (41~71)	45 (20~71)	0.001
BMI, kg/m^2^ (range)	22.2 (16.3~31.5)	23.5 (19.2~29.0)	22.4 (16.3~31.5)	0.283 *
CA-125, U/mL (range)	15.3 (2–74)	181.3 (3–672)	43.4 (2–672)	0.005
parity				0.012
No	20 (40.8%)	0 (0%)	20 (33.9%)	
Yes	29 (59.2%)	10 (100%)	39 (66.1%)	
Menopause				0.054
No	37 (75.5%)	4 (40%)	41 (69.5%)	
Yes	12 (24.5%)	6 (60%)	18 (30.5%)	
Family history of ovarian/breast cancer				>0.999
No	47 (95.9%)	10 (100%)	57 (96.6%)	
Yes	2 (4.1%)	0 (0%)	2 (3.4%)	
US findings				
Laterality of tumor				0.047
Unilateral	40 (81.6%)	5 (50%)	45 (76.3%)	
Bilateral	9 (18.4%)	5 (50%)	14 (23.7%)	
Maximum diameter of lesion, mm(range)	63.6 (17.0–200.0)	75.8 (27.0–168.0)	65.5 (17.0–200.0)	0.322
Maximum diameter of largest solid, mm (range)	10.1 (0–86)	45.7 (0–74)	16.2 (0–86)	<0.001
More than 10 cyst locules				0.055
No	46 (93.9%)	7 (70%)	53 (89.8%)	
Yes	3 (6.1%)	3 (30%)	6 (10.2%)	
Number of papillary projection				<0.001
0	41 (83.7%)	1 (10%)	42 (71.2%)	
1	3 (6.1%)	2 (20%)	5 (8.5%)	
2	0 (0%)	1 (10%)	1 (1.7%)	
3	1 (2%)	0 (0%)	1 (1.7%)	
>3	4 (8.2%)	6 (60%)	10 (16.9%)	
Acoustic shadow				1.000
No	40 (81.6%)	9 (90%)	49 (83.1%)	
Yes	9 (18.4%)	1 (10%)	10 (16.9%)	
Ascites				0.002
No	48 (98%)	6 (60%)	54 (91.5%)	
Yes	1 (2%)	4 (40%)	5 (8.5%)	
B-mode				NA
Unilocular	24 (49%)	1 (10%)	25 (42.4%)	
Multilocular	14 (28.6%)	1 (10%)	15 (25.4%)	
Unilocular-solid	3 (6.1%)	2 (20%)	5 (8.5%)	
Multilocular-solid	8 (16.3%)	3 (30%)	11 (18.7%)	
Solid	0 (0%)	3 (30%)	3 (5%)	
Color doppler				NA
0	41 (83.7%)	2 (20%)	43 (72.9%)	
1	7 (14.3%)	2 (20%)	9 (15.2%)	
2	1 (2%)	5 (50%)	6 (10.2%)	
3	0 (0%)	1 (10%)	1 (1.7%)	

* This *p*-value was calculated by Student’s *t*-test and other *p*-values were calculated by the Wilcoxon rank sum test. yr, year; NA, not available.

**Table 2 jcm-09-02010-t002:** Histologic diagnoses of surgical group (*n* = 54).

	Total (%)
Benign	44 (81.4)
Endometrioma	18 (33.3)
Fibroma	1 (1.9)
Simple cyst	4 (7.4)
Mature cystic teratoma	8 (14.8)
Mucinous cystadenofibroma	2 (3.7)
Mucinous cystadenoma	1 (1.9)
Paratubal cyst	1 (1.9)
Serous cystadenoma	7 (12.8)
Serous cystadenofibroma	2 (3.7)
Borderline and malignancy	10 (18.6)
Mucinous borderline	2 (3.7)
High-grade serous carcinoma	3 (5.4)
High-grade neuroendocrine carcinoma	1 (1.9)
Low-grade endometrioid carcinoma	1 (1.9)
High-grade endometrioid carcinoma	1 (1.9)
High-grade seromucinous carcinoma	1 (1.9)
Poorly differentiated carcinoma	1 (1.9)

**Table 3 jcm-09-02010-t003:** Diagnostic performance of the IOTA (international ovarian tumor analysis)-ADNEX model at each cut-off point of overall malignancy risk.

Cut-off Point	Sensitivity	Specificity	PPV	NPV	LR+	LR-	Accuracy	AUC
5%	0.9	0.755	0.429	0.974	3.680	0.132	0.780	0.828
10%	0.9	0.816	0.500	0.976	4.900	0.123	0.831	0.858
15%	0.9	0.837	0.529	0.976	5.513	0.120	0.848	0.868
47.3% *	0.9	0.980	0.900	0.980	44.100	0.102	0.966	0.940

PPV: positive predictive value; NPV: negative predictive value; LR+: positive likelihood ratio; LR-: negative likelihood ratio; AUC: area under the curve. * 47.3% is an optimal cut-off value that was calculated using the Youden index.

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
