# Peer review of "Validation of IOTA-ADNEX Model in Discriminating Characteristics of Adnexal Masses: A Comparison with Subjective Assessment"

_jcm, 2020, doi:10.3390/jcm9062010_

Round 1
Reviewer 1 Report
This paper evaluates the performance of the IOTA-ADNEX model (based on age, serum CA-125, type of clinic centre, and six sonographic variables), in comparison to the expert opinion of a gynaecologic radiologist, in the assessment and prediction of malignancy in women with ovarian masses. In a cohort of 59 women who were undergoing transvaginal ultrasound for the investigation of ovarian masses at a single institute in Korea, 54 had surgery, and 10 malignant (including borderline) tumours were identified on histology. The investigators found that the IOTA-ADNEX model was highly predictive of malignancy (AUC = 0.924), with a similar performance to the expert gynaecologic radiologist (AUC=0.953).
General comments
The clinical subject of this study is important, given the poor prognosis of ovarian cancer, and the imperfect performance of current methods for differentiating ovarian cancers from the much more common benign ovarian lesions. This paper provides further validation of the previously-developed IOTA-ADNEX model in a new population. The sample size is relatively modest, but the methodology seems reasonable, and the results are quite clearly described. There are a few places where the English language would benefit from editing for style and clarity. Some more specific comments and suggestions follow – these are detailed, but I think all can be quite easily dealt with by the authors.
Specific comments
- Abstract: In the Abstract, the authors give the specificity of the ADNEX model, but not the cut-off point used, or the sensitivity (though these appear later in the main text of the paper). Similarly, the calculated optimal cut-off point is given with specificity but not sensitivity. Was there a particular reason for this (other than perhaps word-count limits)? Also – if the optimal cut-off point calculation is referred to in the abstract, then the methods used to calculate it should perhaps be described in the Methods section, and the result quoted in the Results section (at the moment, I think it just appears in the Discussion section).
- Results, lines 117-118: Power calculation. The authors state ‘The number of participants for screening was determined with the statistical power of 0.8 (p=0.05).’ This is fine, but does not provide any information on the more challenging assumptions for a power calculation (e.g. what difference you are powered to detect). This would be helpful for readers assessing whether the study is adequately powered.
- Table 1: For the continuous variables (e.g. age, BMI, CA-125, ‘Maximum diameter of lesion, mm (range)’ and ‘Maximum diameter of largest solid [component?], mm (range)’), it is not entirely clear whether the number given before the range is a mean or a median. It may be a mix, based on the text (which quotes medians for age and CA-125, but means for diameters of the mass and solid components), but should in any case be specified in the table.
- Results, page 5, Line 125: The text says, ‘The mean mass diameter was 65.6 mm (17-200 mm) and that of the solid component were 16.2 mm (0-86mm).’ Comparing these numbers to those given in Table 1 is confusing, as the [average?] maximum lesion diameter overall is given as ‘55.0 (17.0-200.0)’, and that for the solid component is given as ‘0(0-86)’. It may be that this apparent inconsistency is due to giving median values in Table 1 and means in the text? I think I would favour being consistent - give either a mean or median in the table (presumably based on whether the variable is roughly normally-distributed or not), and then quote the same numbers in the text.
- Results, page 5, lines 132-135 and Table 2: The text here is a little confusing, as it starts by giving percentages for the whole cohort (59 participants), but then moves on to give percentages for the surgical group (n=54). This is fine, but it would be helpful to make it clear when you are moving from one to the other.
- Results, page 5, lines 139-140: I am uncertain as to the meaning of the sentence “Also, it was confirmed by applying z-test to see if there were differences due to the use of ultrasonographic devices (p<0.05).”
- Discussion, page 6, lines 160-163: The authors discuss using the Youden index method to find an optimal cut-off value. I would suggest perhaps explaining how this ‘optimal’ cut-off point is determined, and/or citing a paper that explains the Youden index method, for readers who may be unfamiliar with it. The authors give new results here (the optimal cut-off value that they calculated in the surgical group, with accompanying specificity); I think it would be helpful to indicate the methods used to calculate this (either here or in the Methods section); might it also be helpful to give the accompanying sensitivity? The authors note that the optimal cut-off point they calculated is much higher than that from the original paper (10%) – do you have any thoughts on why your data have led to such different values? Might your higher cut-point lead to less sensitivity, alongside the higher specificity?
- Discussion: The authors rightly point out that both the small number of participants and the short follow-up duration for the non-surgical group are important limitations of this study. Another limitation they do not discuss in terms of wider use of the model is that, as I understand it, the model requires the input of sonographic parameters, which will presumably be dependent on the competence/ experience of the sonographer. While this is probably unavoidable with any ultrasound-based approach, it is an important consideration for wider use of the model. The model may have potential as an aid for a sonographer/ radiologist, but cannot function without them. This will presumably lead to some inevitable variation in the predictive power of the model, depending on sonographer skill. [Though this is obviously a limitation of the model, not of this specific study].
Author Response
Responses to R#1 Comments and Suggestions
General comments
The clinical subject of this study is important, given the poor prognosis of ovarian cancer, and the imperfect performance of current methods for differentiating ovarian cancers from the much more common benign ovarian lesions. This paper provides further validation of the previously-developed IOTA-ADNEX model in a new population. The sample size is relatively modest, but the methodology seems reasonable, and the results are quite clearly described. There are a few places where the English language would benefit from editing for style and clarity. Some more specific comments and suggestions follow – these are detailed, but I think all can be quite easily dealt with by the authors.
Response: Thank you for your invaluable comments. We will do our best in revising our manuscript according to your comments. Actually, we are not English-native speaker, but also we have no native speaker in our OBGY and radiologic departments. So, our manuscript was sent to English-editing service “Enago” prior to the original submission.
Specific comments
Abstract: In the Abstract, the authors give the specificity of the ADNEX model, but not the cut-off point used, or the sensitivity (though these appear later in the main text of the paper). Similarly, the calculated optimal cut-off point is given with specificity but not sensitivity. Was there a particular reason for this (other than perhaps word-count limits)? Also – if the optimal cut-off point calculation is referred to in the abstract, then the methods used to calculate it should perhaps be described in the Methods section, and the result quoted in the Results section (at the moment, I think it just appears in the Discussion section).
Response: We agree with your comments. Initially, we suggested the cut-off point and sensitivity. However, as the number of our cancer cases was relatively small, we used specificity which was so high to excluded almost all benign cases. Because of the word count limitation, we will add the cut-off value and sensitivity in the methods and results sections.
Results, lines 117-118: Power calculation. The authors state ‘The number of participants for screening was determined with the statistical power of 0.8 (p=0.05).’ This is fine, but does not provide any information on the more challenging assumptions for a power calculation (e.g. what difference you are powered to detect). This would be helpful for readers assessing whether the study is adequately powered.
Response: We agree with your comments. We will add more detailed information on why we determined the screening number of participants in ‘Patient selection and study design’.
Table 1: For the continuous variables (e.g. age, BMI, CA-125, ‘Maximum diameter of lesion, mm (range)’ and ‘Maximum diameter of largest solid [component?], mm (range)’), it is not entirely clear whether the number given before the range is a mean or a median. It may be a mix, based on the text (which quotes medians for age and CA-125, but means for diameters of the mass and solid components), but should in any case be specified in the table.
Response: We are sorry to confuse you. We think all continuous variables need a mean because of normally distributed. We will correct a mean in front of them in the ‘statistical analysis’ and ‘results’ section and ‘table 1’.
Results, page 5, Line 125: The text says, ‘The mean mass diameter was 65.6 mm (17-200 mm) and that of the solid component were 16.2 mm (0-86mm).’ Comparing these numbers to those given in Table 1 is confusing, as the [average?] maximum lesion diameter overall is given as ‘55.0 (17.0-200.0)’, and that for the solid component is given as ‘0(0-86)’. It may be that this apparent inconsistency is due to giving median values in Table 1 and means in the text? I think I would favour being consistent - give either a mean or median in the table (presumably based on whether the variable is roughly normally-distributed or not), and then quote the same numbers in the text.
Response: We are sorry to confuse you. One was a median and the other was a mean. We will replace median with mean because all continuous variables were normally distributed.
Results, page 5, lines 132-135 and Table 2: The text here is a little confusing, as it starts by giving percentages for the whole cohort (59 participants), but then moves on to give percentages for the surgical group (n=54). This is fine, but it would be helpful to make it clear when you are moving from one to the other.
Response: Definitely, we agree with you. We will revise the statement to avoid confusion.
Results, page 5, lines 139-140: I am uncertain as to the meaning of the sentence “Also, it was confirmed by applying z-test to see if there were differences due to the use of ultrasonographic devices (p<0.05).”
Response: We agree with you. We did not provide sufficient information on the sentence. A couple of sentences will be added for better understanding.
Discussion, page 6, lines 160-163: The authors discuss using the Youden index method to find an optimal cut-off value. I would suggest perhaps explaining how this ‘optimal’ cut-off point is determined, and/or citing a paper that explains the Youden index method, for readers who may be unfamiliar with it. The authors give new results here (the optimal cut-off value that they calculated in the surgical group, with accompanying specificity); I think it would be helpful to indicate the methods used to calculate this (either here or in the Methods section); might it also be helpful to give the accompanying sensitivity? The authors note that the optimal cut-off point they calculated is much higher than that from the original paper (10%) – do you have any thoughts on why your data have led to such different values? Might your higher cut-point lead to less sensitivity, alongside the higher specificity?
Response: We agree with you. We will add several sentences in the methods and results section according to your comments.
Discussion: The authors rightly point out that both the small number of participants and the short follow-up duration for the non-surgical group are important limitations of this study. Another limitation they do not discuss in terms of wider use of the model is that, as I understand it, the model requires the input of sonographic parameters, which will presumably be dependent on the competence/ experience of the sonographer. While this is probably unavoidable with any ultrasound-based approach, it is an important consideration for wider use of the model. The model may have potential as an aid for a sonographer or a radiologist but cannot function without them. This will presumably lead to some inevitable variation in the predictive power of the model, depending on sonographer skill. [Though this is obviously a limitation of the model, not of this specific study].
Response: Definitely, we agree with you. We will add your comment as one of limitation. However, even if our radiologist was not an early adaptor to new device or software, he did not take a longtime to be used to the input. He got to know it just after some references were read.
Reviewer 2 Report
The accuracy of preoperative diagnosis of adnexal masses is pivotal to improve care as it allows subsequent correct treatment decisions. Ultrasound has a central role in the diagnostic process of an adnexal mass. Therefore the topic of this study is still of great importance.
The ADNEX model developed by the International Ovarian Tumor Analysis (IOTA) group has already been demonstrated to be one of the best model of clinical utility to distinguish between benign and malignant adnexal masses before surgery. This model is offered free to use on the IOTA website, an app can also be downloaded on one’s device and several ultrasound machines offer this software in order to help sonographers.
Soo Young Jeong and colleagues present a very small population of 59 patients, 54 of them underwent surgery, all examined by a single expert sonographer.
Among the 49 patients with an ovarian mass classified as benign, 5 of them didn’t opt for surgery and where followed up with CT scan at only 3-4 months later. Why these patients weren’t re-scanned with ultrasound?
The good accuracy and performance of the ADNEX model has already been demonstrated and validated internally and externally in several larger prospective studies (> 600 patients) with different cut-off vales of malignancy risks and also when used by ultrasound examiners of different expertise.It's also important to remind that the use of ADNEX model needs a good knowledge of the IOTA terminology.
Probably, it could be of more interest a larger prospective comparison among sonographers of different expertise and ADNEX in discriminating between benign and the different categories of malignant masses, not just benign and malignant. The ability of applying ADNEX successfully in triaging ovarian masses, even by non-expert ultrasonographer, could help in avoiding unnecessary surgeries and choosing the right treatments, especially in pre-menopausal women with smaller lesions and who might search for pregnancies in their future.
Author Response
Responses to R#2 Comments and Suggestions
The accuracy of preoperative diagnosis of adnexal masses is pivotal to improve care as it allows subsequent correct treatment decisions. Ultrasound has a central role in the diagnostic process of an adnexal mass. Therefore, the topic of this study is still of great importance.
The ADNEX model developed by the International Ovarian Tumor Analysis (IOTA) group has already been demonstrated to be one of the best model of clinical utility to distinguish between benign and malignant adnexal masses before surgery. This model is offered free to use on the IOTA website, an app can also be downloaded on one’s device and several ultrasound machines offer this software in order to help sonographers.
Soo Young Jeong and colleagues present a very small population of 59 patients, 54 of them underwent surgery, all examined by a single expert sonographer.
Among the 49 patients with an ovarian mass classified as benign, 5 of them didn’t opt for surgery and where followed up with CT scan at only 3-4 months later. Why these patients weren’t re-scanned with ultrasound?
Response: Thank you for your comment. We have already added the short follow up period as one of limitation. CT is known to be a more objective imaging modality than US. So, initial assessment was made with US, but follow up was performed with CT to avoid misdiagnosis made by US. If the participants had been re-scanned with US, discordant cases would have required another follow up or imaging examination (CT or MRI) to determine whether they were really benign.
The good accuracy and performance of the ADNEX model has already been demonstrated and validated internally and externally in several larger prospective studies (> 600 patients) with different cut-off vales of malignancy risks and also when used by ultrasound examiners of different expertise. It's also important to remind that the use of ADNEX model needs a good knowledge of the IOTA terminology.
Response: Definitely, we agree with you. However, our prospective study focuses on validating the previous optimal cut-off value and comparing this model with GY US expert assessment in terms of diagnostic accuracy. Our study demonstrated the optimal cut off value 47.3% was higher than the original value 10%. Moreover, our results suggested that this model has a potential to provide a good diagnostic performance to GY US beginner. These points look more outstanding in our study.
Probably, it could be of more interest a larger prospective comparison among sonographers of different expertise and ADNEX in discriminating between benign and the different categories of malignant masses, not just benign and malignant. The ability of applying ADNEX successfully in triaging ovarian masses, even by non-expert ultrasonographer, could help in avoiding unnecessary surgeries and choosing the right treatments, especially in pre-menopausal women with smaller lesions and who might search for pregnancies in their future.
Response: We agree with you. We need further investigations to determine whether there is any difference regarding the different expertise and differentiation between benign tumor and different categories of malignancy. We will add it as one of limitations.
Reviewer 3 Report
The objective of the study is interesting however the number of subjects needed seems too small to answer the question.
The inclusion period was very short. It would have been more interesting to extend the recruitment period and increase the number of subjects.
Author Response
Responses to R#3 Comments and Suggestions
The objective of the study is interesting however the number of subjects needed seems too small to answer the question.
Response: Thank you for your comments. We understand what you mean. So, we have already added it one of limitations. However, the number of cases were determined by a biostatistician. The number of participants for screening was determined with the statistical power of 0.8 (p=0.05).
The inclusion period was very short. It would have been more interesting to extend the recruitment period and increase the number of subjects.
Response: Thank you for your comments. Definitely, we agree with your comments. Our study was funded by Samsung Medison. The budget was so limited and we was not able to include more participants or to have a longer inclusion period.
Round 2
Reviewer 3 Report
Corrections have been made.